# DDX3 Upregulates Hydrogen Peroxide-Induced Melanogenesis in Sk-Mel-2 Human Melanoma Cells

**DOI:** 10.3390/molecules27207010

**Published:** 2022-10-18

**Authors:** Sanung Eom, Shinhui Lee, Jiwon Lee, Hye Duck Yeom, Seong-Gene Lee, Junho Lee

**Affiliations:** 1Department of Biotechnology, Chonnam National University, Gwangju 61886, Korea; 2GoPath Laboratories, Buffalo Grove, IL 60089, USA

**Keywords:** DDX3, hydroperoxide, melanogenesis, SK-Mel-2 human cancer melanoma cell

## Abstract

DDX3 is a DEAD-box RNA helicase with diverse biological functions through multicellular pathways. The objective of this study was to investigate the role of DDX3 in regulating melanogenesis by the exploring signaling pathways involved. Various concentrations of hydrogen peroxide were used to induce melanogenesis in SK-Mel-2 human melanoma cells. Melanin content assays, tyrosinase activity analysis, and Western blot analysis were performed to determine how DDX3 was involved in melanogenesis. Transient transfection was performed to overexpress or silence DDX3 genes. Immunoprecipitation was performed using an antityrosinase antibody. Based on the results of the cell viability test, melanin content, and activity of tyrosinase, a key melanogenesis enzyme, in SK-Mel-2 human melanoma cells, hydrogen peroxide at 0.1 mM was chosen to induce melanogenesis. Treatment with H_2_O_2_ notably increased the promoter activity of DDX3. After treatment with hydroperoxide for 4 h, melanin content and tyrosinase activity peaked in DDX3-transfected cells. Overexpression of DDX3 increased melanin content and tyrosinase expression under oxidative stress induced by H_2_O_2_. DDX3 co-immunoprecipitated with tyrosinase, a melanogenesis enzyme. The interaction between DDX3 and tyrosinase was strongly increased under oxidative stress. DDX3 could increase melanogenesis under the H_2_O_2_-treated condition. Thus, targeting DDX3 could be a novel strategy to develop molecular therapy for skin diseases.

## 1. Introduction

Melanocytes are melanin-producing cells located in the bottom layer (the stratum basale) of the skin’s epidermis. Melanin [1], a pigment in the melanosome, is primarily responsible for skin color. Besides defining an important human phenotypic trait, melanin has a critical role in photoprotection due to its ability to absorb ultraviolet radiation (UVR) [2,3]. The Fitzpatrick system is the most commonly used system to distinguish different skin pigmentation phenotypes. It characterizes six phototypes (I-VI) by grading erythema and acquired pigmentation after exposure to UVR. Constitutive skin pigmentation as a genetically determined color in the absence of any external factor such as sun exposure can be affected by various regulatory factors [4].

Melanin is synthesized by tyrosinase, a copper-containing metalloglycoprotein and a rate-limiting enzyme that is capable of utilizing L-tyrosine, dihydroxyphenylalanine (L-DOPA), and 5,6-dihydroxyindole as substrates. Other enzymes, including the tyrosinase-related proteins (TRP-1) and dopachrome tautomerase, also known as TRP-2, are also responsible for melanogenesis. Melanogenesis occurs via enzymatic conversion of amino acid tyrosine to melanin pigments through a series of intermediates [5]. Firstly, L-tyrosine is hydroxylated to form L-DOPA. Subsequently, L-DOPA is oxidized to L-dopaquinone, which is further processed into either eumelanin (black or brown pigment) or pheomelanin (yellow or red pigment) [6]. Dopaquinone generally forms eumelanin through spontaneous reactions involving cyclization, decarboxylation, oxidation, and polymerization. TRP-2 can generate 5,6-dihydroxyindole-2-carboxylic acid (DHICA) from dopachrome. TRP-1 catalyzes the oxidation of DHICA to indole-5,6-quinone carboxylic acid. In the absence of thiols, dopaquinone is immediately converted to dopachrome, leading to eumelanin production. However, when glutathione (GSH) and cysteine are present, they can react with dopaquinone intermediates to divert melanin pigment synthesis from eumelanin to pheomelanin through cysteinyl DOPA [7]. Besides these enzymatic reactions, the melanogenic pathway also involves non-enzymatic reactions involving o-quinones generated enzymatically by the action of tyrosinase to produce several unstable intermediates that can polymerize to form melanin. A series of both enzymatic and nonenzymatic reactions in the synthesis of eumelanin and pheomelanin subsequently result in H_2_O_2_ formation [8]. Intrinsic factors such as the inflammatory, endocrine, and central nervous systems, keratinocytes, and fibroblasts can interact with normal skin melanocytes. Extrinsic factors such as drug and ultraviolet radiation also play an important role in regulating its activity [6,9]. Once synthesized, melanin is contained in a special organelle called a melanosome that moves along arm-like structures called dendrites to reach keratinocytes.

DEAD-box DDX3 (DDX3X) has all the properties of an RNA-helicase family, including RNA unwinding activity, transition factor activity, adenosine triphosphate activity, or presence of promoters [10]. DDX3 participates in various cell processes, including cycle progression, cell division, innate immune response, viral replication, and tumor development. DDX3 also has a variety of functions throughout the lifecycle of various viruses. DDX3 mutation can prevent replication of human immunodeficiency virus (HIV), hepatitis C virus (HCV), and other viruses [11,12]. DDX, on the other hand, exhibits antiviral effects on dengue and hepatitis B viruses through interferon stimulation of beta generation. The role of DDX3 in different types of cancer is somewhat controversial [13]. DDX3 acts as a tumor gene in one type of cancer, but shows different forms of tumor suppressant properties [14,15,16]. Human DDX3 helicase is now considered a new attractive target for new drug development.

The oxidative stress could interact with melanogenesis, probably because the photogeneration of ROS reactive oxidative stress could be led by eumelanin and pheomelanin induced by a potential source of H_2_O_2_ [17,18]. H_2_O_2_ and other reactive oxygen species also play an important role in regulating many intracellular pathways, for example, hydroxyl radicals and superoxide radical (O_2_ •−) in melanocytes [19,20]. Sarangarajana et al. have presented that H_2_O_2_ level is directly proportion to the synthesis of melanin in normal melanocytes. Antioxidant or pro-oxidant abilities of H_2_O_2_ depend on the redox state of melanocytes [21]. H_2_O_2_ or NO and oxidative damage stimulating the α-MSH/MC1R or MITF signaling pathway and leading to melanogenesis could mediate the increase in melanin production [22,23]. In melanin synthesis in mammalian skin, H_2_O_2_ is a by-product which is produced following UV irradiation, interacting with chromophores and melanin [24,25]. In a previous study, we have analyzed effects of H_2_O_2_ on melanogenesis using human melanoma SK-Mel-2 cells and mouse melanoma B16F10 cells by measuring melanin content and analyzing expression levels of melanogenesis-related proteins [26], including cAMP-responsive element binding protein (CREB), microphthalmia-associated transcription factor (MITF), tyrosinase, and phenylalanine hydroxylase (PAH). Results of that study showed that H_2_O_2_ could induce melanogenesis by upregulating PAH and activating cAMP/p-CREB/MITF signaling by increasing intracellular cAMP levels through the induction of ATP5B. Although DDX3 is related to a lot of cellular pathways, the biological function of DDX3 in melanogenesis remains unclear. We have previously reported the molecular mechanism involved in the effect of DDX3 on intrinsic apoptosis in HeLa cells. The objective of the present study was to investigate the novel function of DDX3 in melanogenesis. SK-Mel-2 human melanoma cells were selected as a model and H_2_O_2_ was selected as a melanogenesis-inducing agent to study the biological function of DDX3 in melanogenesis, focusing on signaling pathways.

## 2. Results

### 2.1. H_2_O_2_ Activates Melanogenesis and DDX3 Promoter Activity

It has been reported that H_2_O_2_ can induce the expression of melanogenesis-related genes including CREB, MITF, tyrosinase, and PAH in SK-Mel-2 cells and B16F10 cells [26]. In this study, effects of various concentrations of H_2_O_2_ on cell viability of SK-Mel-2 human melanoma cells were determined. Results are shown in Figure 1A. Based on these results, 0.1 mM H_2_O_2_ was chosen for performing subsequent experiments (F2,46 = 29.3, *p* < 0.0001). Melanin expression level in human SK-Mel-2 cells was obviously increased at 4 hrs after treatment with 0.1 mM H_2_O_2_ (Figure 1B). As confirmed by a one-way repeated measures comparisons post hoc test, exposure to 0.1 mM H_2_O_2_ for 4 h increased tyrosinase activity by 2.53-fold compared to control (*p* < 0.0001). DDX3 is known to have an important role in cell cycle regulation [3,27]. In our previous study, we investigated the effect of DDX3 on intrinsic apoptosis in sanguinarine-treated HeLa cells [28]. Here, the effect of DDX3 on melanogenesis induced by H_2_O_2_ was evaluated. It has been reported that 0.1 mM H_2_O_2_ can activate the melanogenesis process [29,30]. Thus, we first assessed whether H_2_O_2_ could induce DDX3 promoter activity. Cells were transfected with pGL2 basic/DDX3 promoter for 24 h and then treated with H_2_O_2_ transiently (0.1 mM for 4 h) before harvesting cells for a luciferase assay. Results showed that DDX3 promoter activity was gradually increased in 0.1 mM H_2_O_2_-treated cells in a time-dependent manner for 4 h (Figure 1D) (*p* < 0.0001). These results may suggest an association between DDX3 and melanogenesis induced by H_2_O_2_ in human melanoma cells.

### 2.2. DDX3 Induces Melanogenesis in H_2_O_2_-Treated Condition Cells

Tyrosinase is known to be a key enzyme in regulating the mammalian melanin synthesis pathway [31]. Melanogenesis can be induced by diverse signaling pathways through activation of pigment-related proteins such as microphthalmia-associated transcription factor (MITF), tyrosinase (TYR), tyrosine-related protein-1 (TRP-1), and tyrosine-related protein-2 (TRP-2). Among these proteins, only tyrosinase is essential for melanogenesis [32]. To verify the effect of DDX3 on melanogenesis in H_2_O_2_-treated cells, SK-Mel- 2 cells were transiently transfected with shDDX or WTDDX3 for 48 h and then treated with 0.1 mM for H_2_O_2_ for 4 h. Melanin content (Figure 2A) was then measured after harvesting cells. Tyrosinase assay results are shown in Figure 2B. Two-way ANOVA revealed a significant effect (or a non-significant effect) of H_2_O_2_ treatment on melanin content, a significant effect of DDX3 (*p* < 0.0001 and *p* < 0.0001, respectively) and a significant interaction between these factors (*p* < 0.0001). Further, two-way ANOVA revealed a significant effect of H_2_O_2_ treatment on tyrosinase activity, a significant effect of DDX3 (*p* < 0.0001 and *p* < 0.0001, respectively) and a significant interaction between these factors (*p* < 0.0001). Post hoc analysis showed that overexpression of DDX3 increased melanin content (by 1.5 times) and tyrosinase activity (by 1.6 times) after H_2_O_2_ treatment compared to control whereas DDX3 knockdown slightly decreased melanin content and tyrosinase activity compared to the control cells. However, SK-Mel-2 cells with DDX3 knockdown by si-DDX3 did not show much difference in melanin content or tyrosinase activity compared to non-treated H_2_O_2_ cells. These results imply that there is an association between melanogenesis and DDX3 in H_2_O_2_-treated cells (Figure 2A,B).

To confirm the role of DDX3 in melanogenesis under H_2_O_2_ treatment, tyrosinase expression was checked by Western blot assay. The results were analyzed using a two-way ANOVA, which showed a significant effect (or a non-significant effect) of H_2_O_2_ treatment on melanin content, a significant effect of DDX3 (*p* < 0.0001 and *p* < 0.0001, respectively) and a significant interaction between these factors (*p* < 0.0001). Similarly, as revealed by post hoc analysis, tyrosinase expression was enhanced in H_2_O_2_-treated transfected cells (Figure 2C,D). DDX3 was increased 1.2 ± 0.6-fold compared to control in WTDDX3-transfected cells and 0.4 ± 0.6-fold in shDDX3-treated cells. However, expression levels of tyrosinase in WTDDX3 under the H_2_O_2_-treated condition were significantly higher than those in shDDX3-treated cells. These findings suggest that overexpression of DDX3 can induce melanogenesis in H_2_O_2_-treated SK-Mel-2 cells.

### 2.3. DDX3 Binds to Tyrosinase in H_2_O_2_-Treated Condition

The previous experiment did not clarify the mechanism of DDX3 involved in the melanogenesis induced by H_2_O_2_ in SK-Mel-2 cells. We hypothesized that DDX could bind to tyrosinase, the key enzyme of melanogenesis. To test this hypothesis, SK-Mel-2 cells were transiently transfected with WTDDX3 for 48 h and then treated with 0.1 mM H_2_O_2_ for 4 h before harvesting for tyrosinase immune precipitation and Western blotting. Results showed that DDX3 co-immunoprecipitated with tyrosine, the melanogenesis enzyme. Two-way ANOVA revealed a significant effect of H_2_O_2_ treatment on TYR, a significant effect of DDX3 (*p* < 0.0001 and *p* < 0.0001, respectively), and a significant interaction between these factors (*p* < 0.0001). A post hoc analysis confirmed that the impact of DDX3 on TYR was strongly increased under oxidative stress as shown in Figure 3A,B. These results suggest that DDX3 might upregulate melanogenesis in SK-Mel-2 human melanoma cells by binding to tyrosinase after H_2_O_2_ treatment. Thus, we proposed a model for the effect of DDX3 on H_2_O_2_-treated HeLa cells (Figure 4).

## 3. Discussion

DDX3 helicase can be a tumor suppressor in many different cancers, including breast, lung, cervical, colorectal, and pancreatic cancers [10,33,34,35]. Moreover, DDX3 might play different roles in the same type of cancer. For example, a decreased level of DDX3 has been found in hepatocellular carcinoma (HCC) caused by hepatitis B virus (HBV), but not in HCC caused by HCV [36]. DDX3 also plays dual roles in breast cancer [37] and colorectal cancer patients [38]. Up to now, there has been no exact explanation about the dual role of DDX in a variety of cancers. DDX3 is involved in the cell signaling pathway of Wnt/β-catenin. It can affect the Wnt regulation cascade, which is crucial for DDX3′s functions in cancer development [39]. DDX3 also modulates cell adhesion and represses E-cadherin expression, resulting in increased cell migration and thus promoting tumor progression [40]. Different roles of DDX helicases might be associated with mutations in the DDX helicase (as can be exemplified by [41]) or virus infections, particularly HCV or HBV. DDX3 knockdown with short interfering RNA (shRNA) or small molecules can suppress cell motility and reduce the metastatic potential in cancer cells and a mouse model [34]. The localization of DDX3 within the cell might also lead to different DDX3 functions. Usually, DDX3 accumulates in the cytoplasm of the cell. However, DDX3 can also be exported from the nucleus to the cytoplasm during tumor progression. DDX3 helicase is a nucleo-cytoplasmic shuttling protein predominantly localized in the cytoplasm of non-malignant cells. It has been suggested that its localization is altered during cell transformation and that such alteration could even contribute to malignancy [42,43].

Hydrogen peroxide (H_2_O_2_) production due to oxidative stress is associated with apoptosis and melanogenesis in melanocytes [26]. It has been reported that some suppressors such as N-feruloyl serotonin can inhibit H_2_O_2_-induced melanogenesis and apoptosis [44]. DDX3 helicase can be a tumor suppressor in many different cancer types. However, the role of DDX3 in melanogenesis has not been reported yet. This is the first report about the involvement of DDX3 in melanogenesis. Results of this study showed that DDX3 upregulated melanogenesis in H_2_O_2_-treated SK-Mel-2 human melanoma cells. Our results also suggested that DDX3 might promote melanogenesis in H_2_O_2_-treated SK-Mel-2 human melanoma cells by binding to tyrosinase. The identification of DDX3 binding partners indicated the function of DDX3 in melanogenesis and is notably novel for the mechanism regulating pigmentation. DDX3 has been reported to be involved in various biogenesis processes as a potential target for cancer treatment or regulation of apoptosis. Another study has reported that cirsimaritin could stimulate melanogenesis in B16F10 cells and SK-Mel-2 human epidermal melanocytes by upregulating MITF and tyrosinase expression and CREB phosphorylation [45].

This study suggests a novel idea of using DDX3 molecular therapy for skin diseases such as skin cancer or vitiligo disease. However, this study did not clearly elucidate the molecular mechanism. In addition, our results were limited to cell lines and melanogenesis-inducing agents. Along with binding proteins, DDX might promote other functions due to stabilization of cell signaling. DDX3 has a variety of complex functions in gene expression systems, viral transmission, and cancer diseases. It is known that DDX3 determines various and contrasting types of cancer [34]. Correlations between DDX3 protein expression levels and various cancer cells have been reported in several studies. In patients with severe lung cancer, the expression level of DDX3 protein is low. A lower expression level of DDX3 has also been reported in patients with smoking history [34]. In contrast, the expression level of DDX3 protein is high in patients with brain tumors, suggesting that it might be a pathological cause of brain cancer. In liver cancer, the expression of DDX protein is more common in men than in women. At the cellular level, DDX is mainly present in the cytoplasm of cancer cells [46]. With regard to cancer diseases, DDX3 has been found to have a variety of functions when it is combined with other proteins. Its main binding proteins include p53, beta-catenin, and WNT, resulting in various signaling mechanisms. p53 is known as a powerful cancer inhibitory protein that can directly activate DDX3 protein [47,48]. This phenomenon is known to control cell proliferation and stabilization of genetic expression [49,50]. DDX3 can act on CK1-alpha in vertebrates and bind to beta-catenin, thus promoting cancer and activating beta-catenin through multiple signaling processes [38,40,51]. Furthermore, abnormal and non-expression of DDX3 can cause WNT to form polymers, leading to cancer [52,53]. Further studies are needed to determine the biological function of DDX3 in melanogenesis as well as the mechanism involved in the effect of DDX3 in promoting melanogenesis under oxidative stress.

In this study, 0.1 mM hydrogen peroxide was used to induce melanogenesis in SK-Mel-2 human cancer melanoma cells by increasing melanin content and the expression of the melanin-synthesizing gene tyrosinase. Interestingly, H_2_O_2_ treatment also increased the promoter activity of DDX3. Under oxidative stress induced by H_2_O_2_, overexpression of DDX3 increased melanin content and tyrosinase expression. By immunoprecipitation with an antityrosinase antibody, it was confirmed that DDX3 could interact with tyrosinase and that such interaction was increased under oxidative stress. A proposed model for DDX3 upregulation of melanogenesis under the H_2_O_2_-treated conditions was presented. This is the first report about the effect of DDX3 on melanogenesis and its potential for developing a molecular therapy to treat skin diseases. Further studies are needed to fully understand the biological effect of DDX3 on the melanogenesis pathway.

## 4. Materials and Methods

### 4.1. Chemicals and Antibodies

Hydrogen peroxide (Cat. no. 216763, Sigma-Aldrich, St. Louis, MO, USA), L-DOPA (Cat. no. 333786, Sigma-Aldrich, St. Louis, MO, USA), and protease inhibitor cocktail were purchased from Merck KGaA (Darmstadt, Germany). Dulbecco’s modified Eagle’s medium (DMEM), fetal bovine serum (FBS), and penicillin–streptomycin were obtained from Gibco BRL (Eggenstein, Germany). The following antibodies were used to detect gene expression: mouse polyclonal IgG anti-DDX3 (produced in rabbits and confirmed by commercial anti-DDX3 antibodies purchased from Novus Biologicals, Littleton, CO, USA; size 73 kDa), mouse monoclonal IgG anti-tyrosinase (SC56505, size 17 and 35 kDa), mouse monoclonal IgG anti-β-actin (Biovision 3598-100, size 44 kDa), mouse monoclonal IgG anti-Myc tag (Millipore 05–724), and mouse monoclonal IgG anti-C-myc (SPM237) (SC7277, size 67 kDa).

### 4.2. Cell Culture

SK-Mel-2 cells were purchased from the Korean Cell Line Bank (KCLB, Seoul, Korea) and cultured in Dulbecco’s modified Eagle’s medium (DMEM) supplemented with 10% fetal bovine serum (FBS) and 1% penicillin/streptomycin at 37 °C in a humidified atmosphere with 5% CO_2_. Cells suspended in culture medium containing 10% FBS were transferred into a flat-bottom 96-well plate. The number of cells/well was calculated to reach the same confluence for all cell culture dishes: 9 × 103 cells/well for 96-well plates, 2.5 × 105 cells/well for 6-well plates, 7.5 × 105 cells/well for 60 mm dishes, and 2 × 106 for 100 mm dishes. Cells passaged fewer than 10 times were used at about 70–80% confluence. Cells were treated with H_2_O_2_ for 1 h to induce melanogenesis. Cell viability was determined with an established colorimetric method using 2,3-bis(2-methoxy-4-nitro-5-sulfophenyl)-2H-tetrazolium-5-carboxanilide inner salt (XTT) obtained from WelGene (Seoul, Korea).

### 4.3. Melanin Content Measurement

SK-Mel-2 cells were seeded into 6-wells plates and cultured for more than 24 h. After reaching 50–70% confluence, cells were treated H_2_O_2_ at different concentrations with a time course. Cells were then washed with PBS and harvested by trypsinization. Cell pellets were solubilized with 2 N NaOH at 80 °C for 2 h and centrifuged at 12,000 rpm for 10 min at 4 °C. The optical density (OD) of the sample was then measured at 420 nm with an ELISA VersaMax Microplate Reader (Versa Max, Remington). The amount of melanin was normalized to the total protein amount in each sample.

### 4.4. Tyrosinase Activity Assay

The tyrosinase activity assay was carried out as previously described. Briefly, SK-Mel-2 cells were seeded into 6-well plates and cultured for more than 24 h to reach 50–70% confluence before treatment. The number of cells/well was calculated to reach the same confluence of 2.5 × 105 cells/well for 6-well plates. After preparing cell culture, cells were harvested, washed with cold PBS, and then lysed with 450 µL of 50 mM PBS (pH 6.8) containing 1% Triton X-100. The mixture was then frozen in dry ice methanol. A volume of 50 µL 10 mM DOPA was added to each sample followed by incubation at 37 °C for 4 h. Cells were treated with H_2_O_2_ for 1 h to induce melanogenesis. The optical density (OD) of the sample was then measured at 475 nm with an ELISA VersaMax Microplate Reader (Versa Max, Remington).

### 4.5. Luciferase Reporter Assay

To measure reporter gene activation (promoter activity), SK-Mel-2 cells were seeded into 12-well plates at a density of 2 × 105 cells/well and transiently transfected with 0.5 µg of a different expression vector for 24 h (for SK-Mel-2). Cells were co-transfected with 20 ng renilla luciferase expression vector (pRL-TK) to normalize the transfection efficiency. After incubation, cells were lysed with 250 µL reporter lysis buffer (Promega, Madison, WI, USA). Luciferase activity from 50/100 µL lysate was measured using a SpectraMax Luminometer (Molecular Devices, Sunnyvale, CA, USA).

### 4.6. Immunoprecipitation (IP)

After seeding into 100 mm dishes, SK-Mel-2 cells were cultured for more than 48 h to reach a confluence at 70% or more before they were transfected with 15 µg pcDNA3-myc/DDX3 vector for 24 h. Cells then were treated with 0.1 mM H_2_O_2_ for 4 h or 8 h and harvested to perform IP. Cell pellets were obtained after washing with PBS and centrifuging. Cell lysate was then used to quantify protein concentration. Each sample (1.5 mg of protein lysate) was mixed with antityrosinase antibody in a total volume of 500 µL lysis buffer, a complete protease inhibitor cocktail, and 50 µL Pierce™ Protein A/G Sepharose beads (Calbiochem, San Diego, CA, USA). Immunoprecipitated proteins were then subjected to SDS-PAGE followed by immunoblotting using specific antibodies.

### 4.7. Statistical Analysis

For melanogenesis activation experiments, statistically significant differences were identified using one-way ANOVA for repeated measures followed by a multiple comparisons post hoc test. Experiments assessing melanin content, tyrosinase activity, and expression were analyzed using a two-way ANOVA with a between-subject factor of the H_2_O_2_ treatment and a between subject factor of DDX3. Statistical analyses were performed using Origin (Ver 8.5; Origin Lab, Northampton, MA, USA). Data represent the mean ± SEM of at least three individual experiments. Differences between the group means were considered significant when *p* < 0.05.

## Figures and Tables

**Figure 1 molecules-27-07010-f001:**
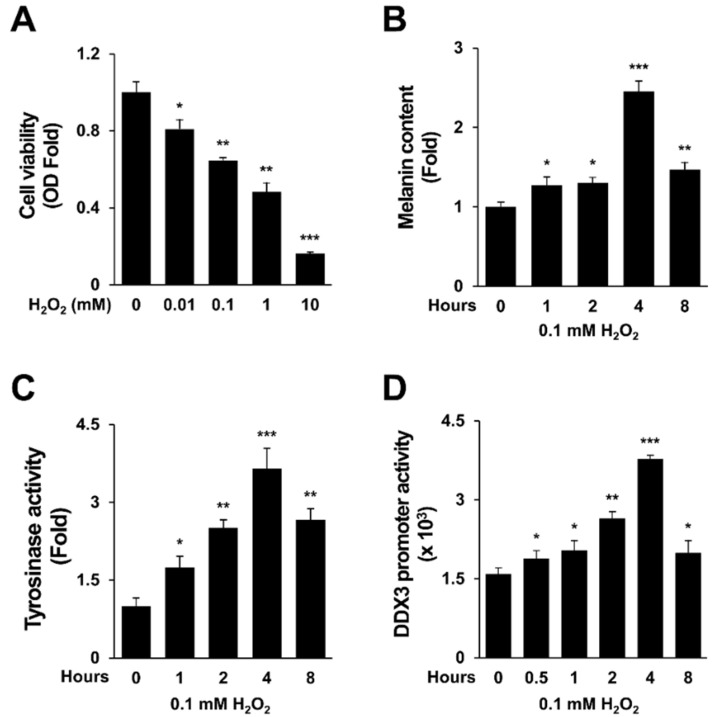
H_2_O_2_ activates melanogenesis and DDX3 promoter activity. SK-Mel-2 cells were treated with various concentrations of H_2_O_2_ for 4 h and then harvested (**A**) or were treated with 0.1 mM H_2_O_2_ for various time courses and then harvested for melanin content assay (**B**) or tyrosinase assay (**C**). Cells were transfected with pGL2 basic/DDX3 promoter for 24 h followed by treatment with 0.1 mM H_2_O_2_ for 4 h before harvesting for luciferase assay (**D**). Data were analyzed using one-way ANOVA with repeated measurements followed by multiple comparisons post hoc test. ***, *p* < 0.05; ****, *p* < 0.001; *****, *p* < 0.0001 compared to control.

**Figure 2 molecules-27-07010-f002:**
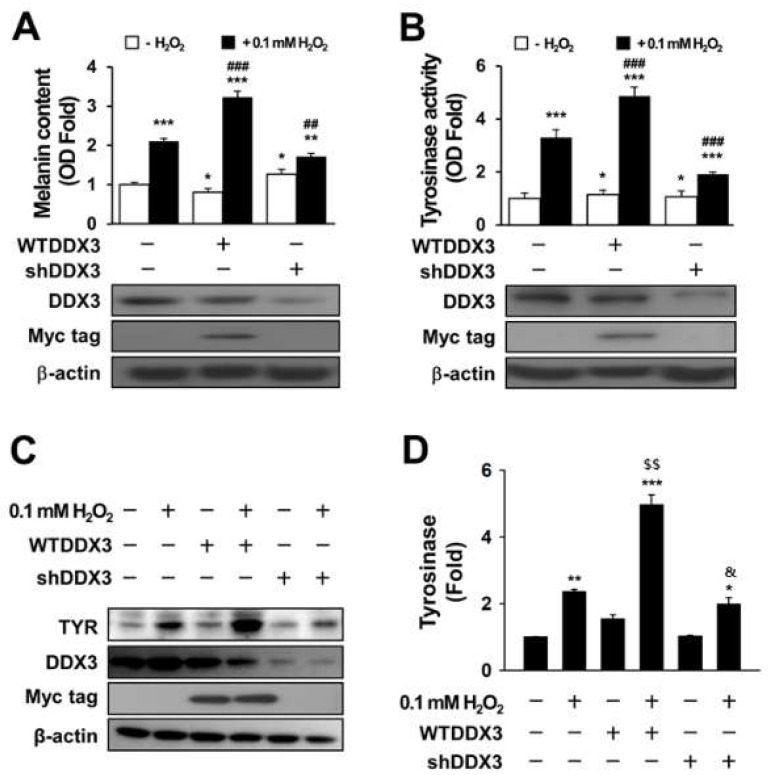
DDX3 induces melanogenesis in H_2_O_2_-treated conditions. SK-Mel-2 cells were transiently transfected with shDDX or WTDDX3 for 48 h and then treated with 0.1 mM H_2_O_2_ for 4 h before harvesting for melanin content assay (**A**), tyrosinase assay (**B**), and Western blot (**C**). The histogram shows relative expression levels of tyrosinase in H_2_O_2_-treated SK-Mel-2 cells with or without DDX (**D**). The data are presented as the mean ± SEM of 4–6 samples from each group. Data were analyzed using two-way ANOVA followed by multiple comparisons post hoc test. ** p* < 0.05, *** p* < 0.001, **** p* < 0.0001 compared to untreated control; *## p* < 0.001, *### p* < 0.0001 indicate differences compared with H_2_O_2_-treated cells after transfection of WTDDX3 or shDDX. *$$ p* < 0.001 indicates differences compared with H_2_O_2_-treated cells after transfection of WTDDX3. *& p* < 0.05 indicates differences compared with H_2_O_2_-treated cells after transfection of shDDX3.

**Figure 3 molecules-27-07010-f003:**
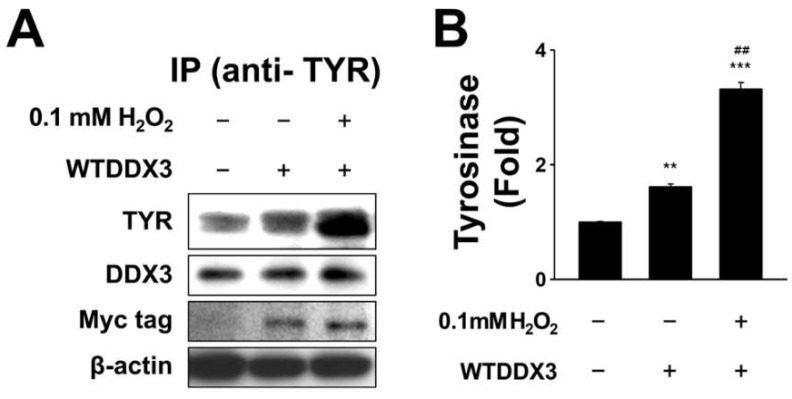
DDX3 binds to tyrosinase in H_2_O_2_-treated conditions. SK-Mel-2 cells were transiently transfected with WTDDX3 for 48 h and then treated with 0.1 mM H_2_O_2_ for 4 h before harvesting for tyrosinase immunoprecipitation and Western blot experiment (**A**). The histogram shows relative expression levels of tyrosinase in H_2_O_2_-treated SK-Mel-2 cells with or without DDX (**B**). Data were analyzed using two-way ANOVA followed by multiple comparisons post hoc test. *** p* < 0.001, **** p* < 0.0001 compared to control cell. *## p* < 0.001 indicates differences compared with H_2_O_2_-treated cells after transfection of WTDDX3.

**Figure 4 molecules-27-07010-f004:**
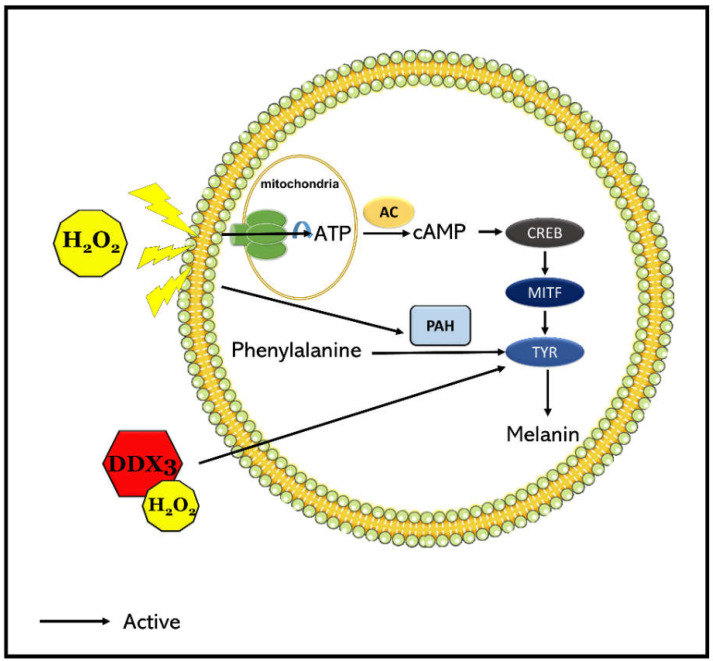
A proposed model of DDX3 for upregulating melanogenesis via binding to tyrosinase in H_2_O_2_-treated SK-Mel-2 cells.

## Data Availability

Data are contained within the article.

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
