# Peer review of "DDX3 Upregulates Hydrogen Peroxide-Induced Melanogenesis in Sk-Mel-2 Human Melanoma Cells"

_molecules, 2022, doi:10.3390/molecules27207010_

Round 1

Reviewer 1 Report

It is a well-structured and documented work, up-to-date for medical oncology, opening the way to the development of new therapeutic targets both in the treatment of cancer and in modeling the oxidative stress produced by its evolution. The involvement of DDX 3 in the modulation of melanin synthesis and free radical response remains a subject of extensive research.

As a recommendation, I think it would be necessary to introduce a section of conclusions to summarize the main results obtained. It was a pleasure to read this work and I wish the authors continued success.

Author Response

It is a well-structured and documented work, up-to-date for medical oncology, opening the way to the development of new therapeutic targets both in the treatment of cancer and in modeling the oxidative stress produced by its evolution. The involvement of DDX 3 in the modulation of melanin synthesis and free radical response remains a subject of extensive research.

As a recommendation, I think it would be necessary to introduce a section of conclusions to summarize the main results obtained. It was a pleasure to read this work and I wish the authors continued success.

Author’s response:

Thank you for expressing it as a well-structured work and seeing it positively. We revised the manuscript in a better way, thanks to your kindness which comments on the details.

Reviewer 2 Report

The authors demonstrated that H2O2 increased melanin synthesis and DDX3 promoter activity, and that WTDDX3 introduction increased melanin synthesis, tyrosinase activity and tyrosinase protein level, while suppression of DDX3 with shDDX3 caused reduction in melanin synthesis, tyrosinase activity, and slight reduction in tyrosinase protein level. Immunoprecipitation showed that WTDDX introduction and H2O2 stimulation increased tyrosinase protein level, but the DDX3 protein level bound to tyrosinase was not changed or may be slightly increased.

The results are well-demonstrated and contains interesting novel findings. The manuscript is well-written compactly.  The followings are some concerns.

1)    They should show if DDX3 mRNA and protein levels are increased with stimulation with H2O2, in addition to promoter activity.

2)    It seems that the level of DDX3 bound to tyrosinase did not increase in proportion to the increase in tyrosinase protein level induced by WTDDX3 introduction in figure 3. How do they explain this result?

3)    Do they think that by binding to tyrosinase, DDX3 changes tyrosinase function or it just stabilizes the tyrosinase and protect it from degradation? Are there any previous reports which sustain these concepts?

4)    Why did they add TNF-alpha (in addition to H2O2?) to the experiment condition in figure 2? They should indicate clearly that they added TNF-alpha in the figure. Or is it just a mistake in the text?

Author Response

The authors demonstrated that H2O2 increased melanin synthesis and DDX3 promoter activity, and that WTDDX3 introduction increased melanin synthesis, tyrosinase activity and tyrosinase protein level, while suppression of DDX3 with shDDX3 caused reduction in melanin synthesis, tyrosinase activity, and slight reduction in tyrosinase protein level. Immunoprecipitation showed that WTDDX introduction and H2O2 stimulation increased tyrosinase protein level, but the DDX3 protein level bound to tyrosinase was not changed or may be slightly increased.

The results are well-demonstrated and contains interesting novel findings. The manuscript is well-written compactly.  The followings are some concerns.

1)    They should show if DDX3 mRNA and protein levels are increased with stimulation with H2O2, in addition to promoter activity.

Author’s response:   

Thank you for your thorough review and precise observations. The increased DDX3 protein levels were shown in Figure 2C. There is not significant band size, but showed relative difference.

2)    It seems that the level of DDX3 bound to tyrosinase did not increase in proportion to the increase in tyrosinase protein level induced by WTDDX3 introduction in figure 3. How do they explain this result?

Author’s response:   

Figure 3A showed the increased tyrosinase protein level in WTDDX2 transfected cell. And additional H2O2 treatment induced the extra expression of tyrosinase

3)    Do they think that by binding to tyrosinase, DDX3 changes tyrosinase function or it just stabilizes the tyrosinase and protect it from degradation? Are there any previous reports which sustain these concepts?

Author’s response:   

We couldn’t find reports. We expect that DDX regulates the activity of tyrosinease or expression of proteins. It seems that further research is needed on whether DDX directly increases tyrosinase. However, our study suggests that it induces an increase in tyrosinase as a result

4)    Why did they add TNF-alpha (in addition to H2O2?) to the experiment condition in figure 2? They should indicate clearly that they added TNF-alpha in the figure. Or is it just a mistake in the text?

Author’s response:   

We corrected it in line 154.

We revised the manuscript in a better way, thanks to your kindness which comments on the details. We hope this revised manuscript will meet your expectations. All revised parts are highlighted in red.

Reviewer 3 Report

Journal of molecules

Research Article;

The article entitled DDX3 upregulates hydrogen peroxide-induced melanogenesis 2 in SK-Mel-2 human melanoma cells’’. The authors best explain the DDX3 is a DEAD-box RNA helicase with diverse biological functions through multicellular pathways. The study investigate the role of DDX3 in regulating melanogenesis by exploring signaling pathways involved. Melanin content assays, tyrosinase activity analysis, and western blot analysis were performed for the analysis of DDX3 in melanogenesis. Transient transfection was performed to overexpress or silence DDX3 genes. Immunoprecipitation was performed using an anti-tyrosinase antibody. Based on results of cell viability test, melanin content, and activity of tyrosinase, a key melanogenesis enzyme. Treatment with H2O2 notably increased the promoter activity of DDX3. The interaction between DDX3 and tyrosinase was strongly increased under oxidative stress. DDX3 could increase melanogenesis under the H2O2 treated condition. Which show that targeting DDX3 could be a novel strategy to develop molecular therapy for skin diseases.  

I carefully read the manuscript and found it suitable for publication in the journal. I accept this article for possible publication. There are some minor mistakes in the article which should be corrected by the authors. After the correction of all the mistakes and revision, the article could be considered for publication in the prestigious molecules Journal.

Comments for Authors

Ø  Write keywords in alphabetical order

Ø  Section Introduction; Revise it. The authors want to put more latest related citations in the introduction part. must include the citation after 2015.

Ø  Use EndNote or Mendeley software for reference sequences.

Ø  Check grammatically and spelling throughout the manuscript. There are some mistakes.

Ø  It is protein kDa in the figure.

Ø  The author needs to show its effect on apoptosis which will bring the novality in the noval effect

Cite the following references;

v  https://doi.org/10.2174/1871520622666220831124321

v  DOI: 10.1038/onc.2014.190

Author Response

The article entitled “DDX3 upregulates hydrogen peroxide-induced melanogenesis 2 in SK-Mel-2 human melanoma cells’’. The authors best explain the DDX3 is a DEAD-box RNA helicase with diverse biological functions through multicellular pathways. The study investigate the role of DDX3 in regulating melanogenesis by exploring signaling pathways involved. Melanin content assays, tyrosinase activity analysis, and western blot analysis were performed for the analysis of DDX3 in melanogenesis. Transient transfection was performed to overexpress or silence DDX3 genes. Immunoprecipitation was performed using an anti-tyrosinase antibody. Based on results of cell viability test, melanin content, and activity of tyrosinase, a key melanogenesis enzyme. Treatment with H2O2 notably increased the promoter activity of DDX3. The interaction between DDX3 and tyrosinase was strongly increased under oxidative stress. DDX3 could increase melanogenesis under the H2O2 treated condition. Which show that targeting DDX3 could be a novel strategy to develop molecular therapy for skin diseases. 

I carefully read the manuscript and found it suitable for publication in the journal. I accept this article for possible publication. There are some minor mistakes in the article which should be corrected by the authors. After the correction of all the mistakes and revision, the article could be considered for publication in the prestigious molecules Journal.

Author’s response:

Thank you for expressing it and seeing it positively. We revised the manuscript in a better way, thanks to your kindness which comments on the details. We hope this revised manuscript will meet your expectations. All revised parts are highlighted in red.

Comments for Authors

Ø  Write keywords in alphabetical order

We rewrote them in line 27, page 1.

Ø  Section Introduction; Revise it. The authors want to put more latest related citations in the introduction part. must include the citation after 2015.

We corrected it in line 31, page 1.

Ø  Use EndNote or Mendeley software for reference sequences.

We corrected it.

Ø  Check grammatically and spelling throughout the manuscript. There are some mistakes.

I appreciate your sincere comment. The histogram graph of each figure measured the amount of protein and quantified it as an increase multiple of the control group.

Ø  It is protein kDa in the figure.

I appreciate your sincere comment. The histogram graph of each figure measured the amount of protein and quantified it as an increase multiple of the control group.

Ø  The author needs to show its effect on apoptosis which will bring the novality in the noval effect

I added the promotor activity of DDX was increased by oxidative condition, and then it lead to melanin content of melanogenesis by increasing of tyrosinase expression in last paragraph in discussion.

Cite the following references;

v  https://doi.org/10.2174/1871520622666220831124321

v  DOI: 10.1038/onc.2014.190

Thank you for your thorough review and precise observations.

Round 2

Reviewer 2 Report

I think the manuscript has been well revised.